

# `RavenR` v2.1.4: an open source R package to support flexible hydrologic modelling

Robert Chlumsky[1], James R. Craig[1], Simon G.M. Lin[1], Sarah Grass[2], Leland Scantlebury[1,3], Genevieve Brown[4], and Rezgar Arabzadeh[1]

[1]Department of Civil and Environmental Engineering, University of Waterloo, Waterloo, ON, Canada
[2]Geoprocess Research Associates, Edmonton, AB, Canada
[3]Department of Land, Air and Water Resources, University of California, Davis, CA, United States
[4]Northwest Hydraulic Consultants Ltd., North Vancouver, BC, Canada

**Correspondence:** Robert Chlumsky (rchlumsk@uwaterloo.ca)

**Abstract.** In recent decades, advances in the flexibility and complexity of hydrologic models has enhanced their utility in scientific studies and practice alike. However, the increasing complexity of these tools leads to a number of challenges, including steep learning curves for new users and in the reproducibility of modelling studies. Here, we present the `RavenR` package, an R package that leverages the power of scripting to both enhance the usability of the Raven hydrologic modelling framework and provide complimentary analyses that are useful for modellers. The `RavenR` package contains functions that may be useful in each step of the model-building process, particularly for preparing input files and analyzing model outputs, and these tools may be useful even for non-Raven users. The utility of the `RavenR` package is demonstrated with the presentation of six use cases for a model of the Liard River basin in Canada. These use cases provide examples of visually reviewing the model configuration, preparing input files for observation and forcing data, simplifying the model discretization, performing reality checks on the model output, and evaluating the performance of the model. All of the use cases are fully reproducible, with additional reproducible examples of `RavenR` functions included with the package distribution itself. It is anticipated that the `RavenR` package will continue to evolve with the Raven project, and will provide a useful tool to new and experienced users of Raven alike.

## 1 Introduction

Hydrologic models are used for numerous applications, including streamflow prediction, flood forecasting, reservoir level forecasting, and in a scientific capacity to advance our understanding of hydrologic systems. Historically, most hydrologic models have been designed with a fixed model structure comprised of a predefined set of environmental processes, while the input data and model parameters may vary from watershed to watershed. While these fixed model structures (e.g. GR4J, (Perrin et al., 2003)) may provide sufficient performance in some catchments, they are not adequate in all catchments, environments, or hydrologic applications (Hoey et al., 2014). Numerous studies have called this fixed structure paradigm into question, and have instead called for the development of flexible modelling frameworks (Leavesley et al., 2002; Clark et al., 2011; Fenicia et al., 2011), which would allow the modeller to possess more control over the model-building process. This has resulted in





the emergence of flexible modelling frameworks in the literature (e.g., Orellana et al., 2008; Clark et al., 2008; Kavetski and Fenicia, 2011; Clark et al., 2015; Knoben et al., 2019; Craig et al., 2020), and recent studies have been extensively supported

by the use of these frameworks (Pilz et al., 2020; Remmers et al., 2020; Chadalawada et al., 2020; Knoben et al., 2020; Spieler et al., 2020; Mai et al., 2020; Chlumsky et al., 2021c).

     The power contained in these flexible hydrologic models is limited in part by the modeler's ability to take advantage of it. In an ideal setup, a modeller would find converting their system conceptual model to a numerical model a seamless process; in actuality, setting up a numerical model often involves data wrangling and other tedious tasks, with decisions ranging from those

with relatively little impact on the final modelling results (e.g., how to combine dozens of text files) to potentially problematic and highly impactful decisions (e.g., time series interpolation or model structure adjustments). Even among hydrological modelling software that have graphical user interfaces (GUIs), few offer the ability to easily deploy and compare successive model runs, resulting in a potentially large amount of time devoted to relatively trivial tasks, such as organizing model files and comparing successive model runs.

In order to address some of these challenges, new tools must be developed to bridge the gap between complex, customizable tools and the ability for modellers (in particular, new users) to fully understand and deploy these tools. Increasingly, freely available, open-source scripting languages, such as Python and R, are being employed by modellers to create, visualize, and evaluate their models (Jackson et al., 2019; Slater et al., 2019; Astagneau et al., 2021). While the languages themselves can carry significant learning curves, they unlock a wide range of time-saving features due to their ability to reproducibly automate

tasks in concise code that can be re-used between projects. Additionally, these languages can be greatly expanded through the straightforward installation of downloadable packages that allow the performing of complex statistical analyses, obtaining and manipulating data, producing publication-ready plots, and even building interactive web visualizations, to be reduced to a handful of lines of code. Many features and tools that would be inappropriate or inadvisable to build into the source code of hydrologic models (e.g., downloading and quality controlling input data) are perfectly suited to scripting languages.

Reproducibility is essential to scientific advancement: to build a usable body of knowledge, we must be able to trust, rely on, learn from, and when necessary, upend the lessons of past experiments (Hutton et al., 2016). There are studies suggesting that large proportions of scientific studies may not be reproducible (Baker, 2016; Camerer et al., 2018), including the field of computational hydrology (Hutton et al., 2016; Chawanda et al., 2020). Studies deploying proprietary or even free but not open-source software, such as HEC-HMS (Hydrologic Engineering Center, 2020), pose issues in their reproducibility. While

open-source modelling software provides a transparent codification of the model, the reliability of the resulting model depends heavily on the preparation of model input files and required data, which may not necessarily undergo the same level of scrutiny and transparency in a given modelling exercise. The communication of modelling methodology and workflow in science has historically been accomplished through the inclusion of a detailed methods section within research manuscripts. However, in hydrology, the code and data that are needed to generate published results are often not made available, and the complexity of

many hydrologic models and analyses make it infeasible for authors to include all necessary details for full reproducibility in their manuscripts (Hutton et al., 2016). Increasingly, scientists and engineers are being encouraged to use (and in some cases publish) computationally reproducible versions of their analyses (National Academies of Sciences, Engineering, and Medicine,





2019). Scripts, as a perfect record of the data manipulation, model setup, post-processing, and even figure creation steps, are the ideal tools to accomplish this.

R, in particular, has gained significant ground in hydrology, entering the toolbox of many in both consulting and academia (Anderson et al., 2018; Slater et al., 2019; Astagneau et al., 2021). This is due in part to the already robust package ecosystem awaiting hydrologists; the Comprehensive R Archive Network (CRAN), which hosts and tests R packages, allows for easy access to packages for a variety of data processing tasks such as downloading data (`tidyhydat` (Albers, 2017)), examining data (`trend` (Pohlert, 2020)), manipulating shapefiles and spatial data (`sf`, `raster` (Pebesma, 2018; Hijmans et al., 2021)),

evaluating model outputs (`hydroGOF` (Mauricio Zambrano-Bigiarini, 2020)), and visualizing data (`ggplot2` (Wickham, 2016)). Many hydrology-specific open-source packages have been developed in recent years, such as the `CSHShydRology` package (Shook et al., 2021), `Evapotranspiration` (Guo et al., 2020), and many other packages reviewed in the literature (Slater et al., 2019; Astagneau et al., 2021) and listed on CRAN Task View for Hydrology (https://cran.r-project.org/web/views/ Hydrology.html). R is also being used extensively in teaching hydrology to professionals and graduate students, and packages

have also been used directly as educational tools, such as the `airGRteaching` package (Delaigue et al., 2020, 2018).

    Here, we introduce an R package with a collection of tools to aid a modeller in preparing, running, and post-processing results from custom hydrologic models developed with the hydrologic modelling framework Raven. Many of the tools are not solely Raven-specific: functions exist to plot time series, analyze yearly patterns, and compute relevant statistics. However, the package importantly contains a robust suite of functions for creating, reading, and manipulating Raven model files. Specific

attention has been paid to supporting the testing, comparison, and diagnosis of models built with variable model structure; many of these tools are unique. The intended purpose of the `RavenR` package is to enable modelers to simplify, automate, and document their model creation process, effortlessly facilitate model visualization and evaluation, and to expand the flexibility of the Raven hydrological modelling framework through scripting.

## 2   Methods

Section 2.1 briefly discusses Raven, and provides context for the `RavenR` package as a tool that enables improved workflows with Raven. Section 2.2 discusses `RavenR` in more detail, including a typical model-building workflow (Section 2.2.1), the installation and documentation available within `RavenR` (Section 2.2.2), and a description of the sample data sets available within the package and external to the package (Section 2.2.3).

### 2.1   Raven hydrologic modelling framework

Raven is an open-source software framework that can be used to build models from a selection of more than 100 available process algorithms (Craig et al., 2020). It is estimated that at least $8 \times 10^{12}$ different hydrologic model configurations may be setup using Raven (Mai et al., 2020) and this number is continuously increasing as new options are added to the software. Raven is built for flexibility not only in process representation, but also in enabling multiple numerical schemes, discretization schemes, input data types, and in providing the user control over output options. Raven is a fully object-oriented code written



in C++, and is typically run from a command line. The input and output files are generally stored as text files (*.txt or *.csv) or in NetCDF (Network Common Data Format) format. This allows all model files to be stored as non-proprietary formats, and to be read and processed with any number of available tools for manipulating files.

The primary input files required for Raven (listed by file extension) include:

1. *.rvi - primary input file which defines the model structure, timestep, duration, and a number of additional options

2. *.rvp - model parameter specification and soil/vegetation/land class definitions

3. *.rvh - model discretization, including all subbasin and hydrologic response unit (HRU) information

4. *.rvt - time series data, including forcing and observational data; this file often points to other *.rvt files with data sets for various stations and locations

5. *.rvc - initial conditions for the model run

Raven provides complete control over its output generation (Craig et al., 2020), a relatively uncommon feature in hydrologic modelling software. A large proportion of the computational cost of a model run is often used in the writing of output files, and thus a substantial computational burden can be alleviated if only the required outputs are written. This could be a single diagnostic metric for calibration, a single time series, or the complete mass and energy balances of the model for debugging or auditing purposes. Raven also allows for custom outputs to be generated for a given statistical, spatial and temporal specification

and state variable, such as the monthly average of daily snow depth for a particular set of subbasins (Craig et al., 2020).

This flexibility of Raven over the modelling process provides the modeller with a lot of power in configuring and running their hydrologic model, but also provides some challenges in preparing files and working with the many possible outputs. The command-line execution of the program and the lack of a user interface can present a learning curve for new users, but also enables scripting languages to easily interface with Raven, and for Raven to be deployed in high-performance computing

environments.

A number of utilities exist to support the usage of Raven models, including the RavenPy (https://github.com/CSHS-CWRA/RavenPy) for creating, running, and post-processing Raven models within Python, and HydroGlyph (http://raven.uwaterloo.ca/hydroglyph/) for visualizing Raven time series output data. Hydrologic model support is also provided by many model-independent packages, such as the CSHS-hydRology package (Anderson et al., 2018). However, `RavenR` is the most compre-

hensive tool for preparing input files and performing a range of analyses with Raven output files.

## 2.2   `RavenR` software description

### 2.2.1   `RavenR` overview

The `RavenR` package is developed in R, and is a collection of tools to aid the modeller in preparing, running, and post-processing files associated with a hydrologic model developed using Raven (Craig et al., 2020). `RavenR` is not intended to



provide every tool needed to manipulate and analyze Raven input and output files, as the flexibility of the Raven framework would require a vast collection of highly specific scripts to accommodate the needs of all modellers. Instead, the package aims to reduce the effort required to use Raven, and allows the modeller to more effectively use the open-source scripting environment of R in their workflows. This may also reduce the learning curve of Raven that is created by its flexibility, as the package provides the means to guide new users through the generation and manipulation of common files, and reduces the

burden in analyzing the model results. This can be particularly helpful for users migrating from GUI-based software such as HEC-HMS (Hydrologic Engineering Center, 2020).

     The available functions within `RavenR` can be broadly categorized by their utility into the main categories of: 1) preparing input files, 2) reading output files, 3) running Raven, 4) tools for hydrologic analyses, and 5) support utilities (e.g. time series processing, water year analysis, etc.). The typical workflow for `RavenR` is closely related to the workflow required

for the development and use of any hydrologic model, including one developed with Raven. This includes the collecting and processing of data for the model, determining the model structure, creating model input files in the format required by the modelling software, executing the model, and analyzing the results of the model for hydrologic consistency and performance. This can include exercises in model calibration and validation, uncertainty analysis, identifiability analysis, and project-specific simulations or adjustments to the model runs.

The typical workflow for developing a hydrologic model and examples of `RavenR` functions that may be used to support each step are provided in **Table 1**.





**Table 1.** Typical workflow table for building hydrologic models and connection to `RavenR`.

| Step | Activity | Description | RavenR Functions |
|------|----------|-------------|------------------|
| 1 | Collect/Prepare Data | Preparation and quality control of Raven input files (e.g., *.rvi files from template, *.rvt files), often from public data sources | `rvn_rvi_write_template`, `rvn_rvt_tidyhydat`, + 19 others |
| 2 | Discretize Watershed | Quality control of implemented discretization scheme and further simplification (e.g., aggregating very small or similar HRUs) | `rvn_rvh_clean_hrus`, `rvn_subbasin_network_plot` |
| 3 | Identify and Describe Processes | Model structure development and process algorithm selection | `rvn_rvi_connections`, `rvn_rvi_process_diagrammer`, `rvn_rvi_process_ggplot` |
| 4 | Parameterize the Model | Model parameter definition and parameter value specification | `rvn_rvi_get_params` |
| 5 | Execute the Model | Running the Raven (or other hydrologic) model | `rvn_download` & `rvn_run` |
| 6 | Processing Model Outputs | Reading and processing model output files for analysis | `rvn_hyd_read`, `rvn_custom_read`, + 7 others |
| 7 | Plots and Model Diagnostics | Checking model performance with a number of analyses, reality checks, and diagnostics (often in conjunction with model calibration and validation) | `rvn_annual_peak_flow`, `rvn_monthly_vbias`, + 24 others |
| 8 | Report Results | Generating quality graphics and workflows to communicate results | Functions from step 7 + additional R libraries (e.g., `ggplot2` & `rmarkdown`) |



Although the model-building process is listed in **Table 1** as a series of steps, in practice it is not linear, but rather iterative and cyclic. For example, a model diagnostic (step 7) may show that inadequate model performance can be remedied by the inclusion of additional forcing data, requiring new data to be written to file (step 1). It is also recommended or common practice

in modelling to begin with a simpler model and proceed to a more complex one (e.g., Fenicia et al., 2008), which may require iteration on steps 2-6 to potentially modify the structure (e.g. spatial and temporal discretization, hydrologic processes) after a basic model has been established. Model calibration would typically involve an iteration upon steps 4-6 with a calibration algorithm, and a calibration that includes model structure (e.g., Spieler et al. (2020), Chlumsky et al. (2021c)) would effectively iterate upon steps 3-6. The iterative need for these model-building steps emphasizes the benefit of tools (including those in

RavenR) that can reduce the overhead in simple but repetitive tasks, such as producing figures and writing data to a specific file format.

The functions within the RavenR package are named, where appropriate, by the three letter Raven file name or short abbreviation corresponding to the output file that they interact with, e.g., rvn_rvi_connections for processing the *.rvi file structure or rvn_res_read for reading the output ReservoirStages.csv file. Other functions simply use illustrative names to

convey their purpose (e.g. rvn_budyko_plot). This naming convention provides some navigability of the package functions to the new user, even before the package documentation is reviewed (see Section 2.2.2).

The RavenR package has a number of preferred data formats and related package dependencies. Most plots are generated using the ggplot2 (Wickham, 2016) and related libraries from the so-called tidyverse, including dplyr (Wickham et al., 2021a) and tidyr (Wickham, 2021) for data manipulation. This allows all plots to be exported as plot objects and further

manipulated by the user as desired, and removes the need for all plot options to be wrapped into RavenR functions. Time series handling is done through the lubridate (Grolemund and Wickham, 2011) and xts (Ryan and Ulrich, 2020) packages, where the extensible time series (xts) format is generally expected for time series data. Finally, support for network analysis is done through the igraph package (Csardi and Nepusz, 2006), which primarily supports the organization of watershed discretization connections (*.rvh file) and the network of model structure connections (*.rvi file), including the related plot

functions, e.g., rvn_subbasin_network_plot and rvn_rvi_process_diagrammer.

### 2.2.2 Installation and documentation

The package is developed as a free and open-source software tool, which is ideal for maintaining transparency and reproducibility in workflows related to hydrologic modelling and all steps involved. The stable package version is available for download through CRAN (currently version 2.1.4), which can be installed in R using the command install.packages("RavenR").

The development version of the package is available on Github (https://github.com/rchlumsk/RavenR) and may be installed using the devtools library (Wickham et al., 2021b) as devtools::install_github('rchlumsk/RavenR'). Both installation commands resolve the dependencies associated with the package.

The RavenR package is fully documented and contains a description of inputs, outputs, with an usage example for each function consistent with the standards for CRAN packages. In addition to the package documentation, an introductory vignette

*Introduction to RavenR*, is included with the package, which discusses getting started with the package and how it may be used





in a manner that is more useful to new users of Raven and `RavenR`. The introductory vignette is available with the command `install.packages("RavenR")`.

### 2.2.3   Sample data sets

In the interest of reproducible examples, the `RavenR` package contains a number of sample data sets and raw data files
embedded within the package which are used within the function examples. Sample data is embedded directly as imported data (accessible with the `data` function in R) for a number of file output types (hydrograph, watershed storage, etc.), as well as sample data for the `tidyhydat` and `weathercan` packages. The latter sample data allows the function examples to run without a dependency on the mentioned data-retrieval packages. Raw data files are also included (accessible with the `system.file` function in R), which allow for the testing of reading raw data directly. The examples where raw data files are
first read into R using `RavenR` functions may be more helpful than examples which call sample data directly with the `data` command, since the workflow will be closer to the one applied in practice.

The sample Raven output files and data that is distributed with the `RavenR` package were generated from a model of the Nith watershed, which is located immediately west of Kitchener-Waterloo in Ontario, Canada. The Raven model of the Nith watershed can be found in full on the Raven webpage (http://raven.uwaterloo.ca/downloads.html) in the Tutorials 1-4 download
set. Numerous additional Raven models are available from this page, including the model of the Liard River basin (Brown and Craig, 2020), which is used in the `RavenR` case studies in this manuscript (**Section 3**).

## 3   Use cases of the **RavenR** package

In this section, we present a number of use cases of the `RavenR` package. These cases are not intended to be a comprehensive review of all the applications for the `RavenR` package, but to provide the reader with a partial demonstration of how the
package may be used in conjunction with Raven. The cases are discussed in the context of hydrologic modelling with flexible frameworks more broadly, and provide cases and checks that are likely to be useful when deploying both Raven and non-Raven hydrologic models.

The use cases are presented in approximate order of the model-building process (**Table 1**), beginning with the generation of model input files and proceeding to the analysis of output files. These use cases include examples and discussion of almost
all steps of the model-building process, with the exception of steps 4 and 5. Tools for these steps, such as running Raven from within R using `rvn_run` (step 4), exist within the `RavenR` package but are not discussed in detail in this section.

All R code and model files required to generate the results and figures in this section are provided in an open-source Zenodo repository (https://doi.org/10.5281/zenodo.5534818), and utilize the version of `RavenR` currently available on CRAN (version 2.1.4).





## 3.1 Liard River basin

The use cases presented here utilize the Liard River basin model built with Raven. The Liard River basin is located in northern Canada, spanning the Yukon Territory, Northwest Territories, British Columbia and Alberta. The Liard River is the largest tributary to the Mackenzie River, with a total contributing area of $275\,000$ km$^2$ (Brown and Craig, 2020). The basin includes a variety of landforms, including mountainous regions and wetland-dominated regions. There are varying degrees of difficulty when trying to accurately represent these various landforms in a hydrologic model. Additional details on the Liard River basin, and the corresponding hydrologic model developed for the basin with Raven, can be found in Brown and Craig (2020), which also describes the manual calibration process that was undertaken for the model.

## 3.2 Input file processing

An early step in the model-building process is the collection of data and preparation of model input files (step 1 in **Table 1**. While the supporting data analysis may not require expert knowledge of hydrology *per se*, the data preparation can require a substantial amount of time and effort in the modelling process. Further, the reproducibility and merit of research may rest on the ability to access and reproduce the original and processed intermediate data, which is vastly improved by the use of a scripting environment that in effect documents the steps taken to prepare the data files (Anderson et al., 2018). As such, the use of scripting tools, such as those that will be discussed in this section, may be used to both reduce the effort required to prepare input data files and improve the reproducibility of the research or applied project.

This section discusses the preparation of the model structure configuration, the preparation of forcing data and observation data, and modifications to the model discretization file. Additional utilities related to input files exist within the package (such as providing parameter information), but these applications discuss a large proportion of the workflow that would be required in developing a set of Raven input files.

### 3.2.1 Model configuration

One of the key characteristics of Raven is flexibility, including the ability to customize the model structure in terms of the organization of water storage units and selection of process equations (Craig et al., 2020). The hydrologic process schematic is specified through the main Raven input file, in which the number of soil layers, the list of hydrologic processes and the set of 'to' and 'from' compartments, etc., are defined.

For new and even more experienced users of Raven, understanding the model structure and making changes within the primary input file may be a somewhat daunting task, particularly when the model must be initially developed. Fortunately, the Raven User's Manual (Craig, 2020) provides the templates for a number of model structures, which can be used as a starting point for constructing a customized hydrologic model. These are largely based on emulations of published hydrologic models in the literature (e.g. UBCWM, HMETS, etc.), although some are based on research models that have been developed within Raven (e.g. the Canadian Shield model). Once a base model has been selected, components of the model may be modified using the many process options available within Raven which are documented in the Raven User's Manual (Craig, 2020). The





large number of process options available to the user provide no shortage of model structure tweaks to customize their model. A critical step in making these adjustments to model structure is understanding the structure and ensuring that it is consistent with the modeller's conceptual understanding of the system (step 3 in **Table 1)**. While Raven itself does not currently have

a user interface deployed that can visualize the model structure, functions within the `RavenR` package can generate a model schematic from the contents of the model input (*.rvi) file. The ability to visualize this structure can be critical in understanding the model structure and ensuring the conceptual understanding is consistent with the implemented structure. This can also be used to check for state variables or storage units with an improper number of connections, such as a soil layer with no outflow mechanism.

The general workflow within `RavenR` to generate a model *.rvi file and visualize the contents is as follows:

1. A template model structure is selected and written to file using the `rvn_rvi_write_template` function.

2. The file may be manually modified in consultation with the Raven User's Manual (Craig, 2020)

3. The file may be read into R using the `rvn_rvi_read` function

4. The process connections from the file can be processed using the `rvn_rvi_connections` function

5. The process diagram can be generated either in ggplot format using the `rvn_rvi_process_ggplot` function, or as a diagrammer plot using the `rvn_rvi_process_diagrammer` function

6. (optional) the ':CreateRVPTemplate' command can be used to generate a template *.rvp (parameter) file when Raven is executed

7. (optional) the `rvn_rvi_get_params` function may be used to obtain a data frame of parameters, ranges, and default
parameter values for parameters that are included in the hydrologic model, based on the model structure configuration

An example of the process diagram is provided for the Liard River basin in **Figure 1**. From this figure, the directional connections between water storage compartments in the model can easily be ascertained and verified, allowing modelers building a new model and modelers inheriting a model alike to quickly understand the movement of water in their current setup. For instance, in the Liard model we can see the model has capacity for precipitation to enter specific wetland and depression

compartments, snow can melt and refreeze, and fast and slow (upper and lower) reservoirs exist to represent groundwater processes as different time scales (Brown and Craig, 2020). A single-layer topsoil compartment is used to connect the surface water and subsurface domains in the model along with a vadose zone reservoir to help represent a karst structure within the model. We can see that all processes that move water to glacier are conditional based on the HRU type (glacier HRU). Ponded water is moved to depression storage under the condition of being a wetland and surface water only directly evaporates to the

atmosphere if the HRU is a lake. The karst groundwater structure which was implemented in the model is only applicable to a subset of the HRUs which accounts for the conditional connections between SOIL[0], surface water, the vadose reservoir, and the fast and slow reservoirs. Reviewing and verifying these conditional exceptions along with the connections between other state variables can help ensure that the model is appropriately structured.



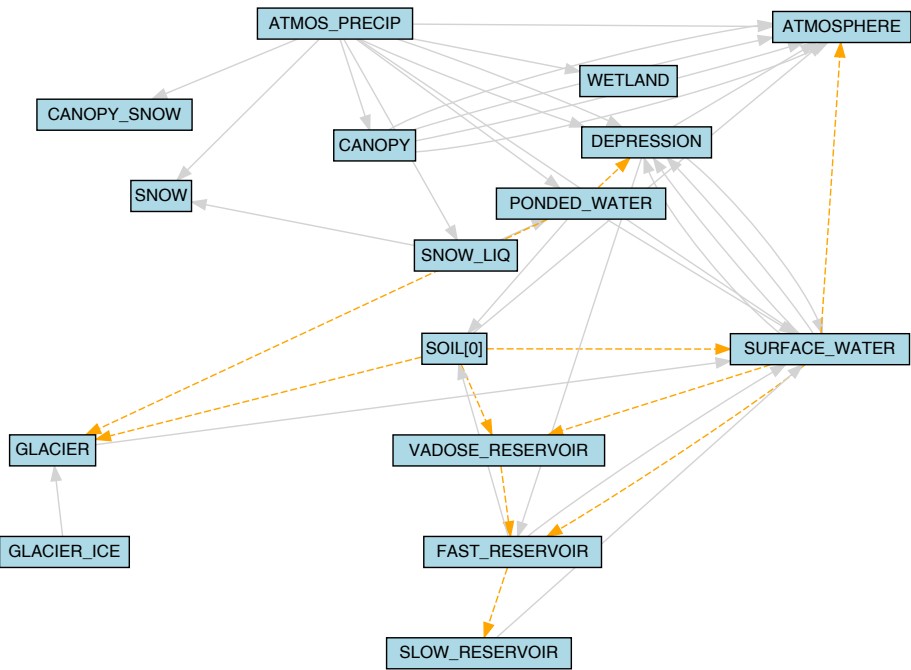

**Figure 1.** The model configuration of the Liard basin, generated from the Liard model *.rvi input file with the `rvn_rvi_process_diagrammer` function. Solid grey lines indicate connections between state variables, and dashed orange lines indicate conditional connections.

Typically, diagrams such as these are arduous to produce for highly-flexible modelling software such as Raven. Here, the
function has been automated to create publication-ready diagrams for most model setups.

### 3.2.2 Forcing data

Meteorological forcings (e.g. precipitation, temperature, wind speed) drive the hydrologic model responses. When not collected as part of a project, these data are often obtained from online, freely available public sources generally collected, processed, and maintained by local and/or public agencies. These data are likely to require some quality control before ingestion into
the model, such as addressing data flags, removing erroneous data, and converting units (step 1 in **Table 1**). This process can be quite tedious, especially when combining multiple data sets of various formats, time steps, and quality. The RavenR package offers the `rvn_rvt_write_met` function for writing forcing data directly to the Raven *.rvt format: the function defaults are configured to accept outputs from the `weathercan` R package, which automatically downloads data for Canadian meteorological stations maintained by Environment Canada (LaZerte and Albers, 2018).

In this use case, daily meteorological data for a 20-year period is downloaded, interpolated, and written to Raven *.rvt format. The `weathercan` R package is used to search for stations within 500 km of Fort Liard and with data records spanning from 1985 to 2005. A subset of stations meeting these criteria is downloaded for pre-processing. Missing values in the meteorological





```
fort_liard <- c(60.241711, -123.467377)
stns <- weathercan::stations_search(coords=fort_liard, interval="day",
                           dist = 500, starts_latest = 1985, ends_earliest = 2006)

weather_dl(stns$station_id[1:10], interval="day",
           start="1985-10-01", end="2005-10-01") %>%
  rvn_met_interpolate(cc=c("max_temp", "min_temp", "total_precip"),
                      key_stn_ids = stns$station_id[1:5]) %>%
    rvn_rvt_write_met()
```

**Figure 2.** Minimum code required for the use case described in **Section 3.2.2** of downloading, interpolating, and writing meteorological data into Raven *.rvt format using the `weathercan` and `RavenR` R packages. The pipe operator (`%>%`) from the `dplyr` package is used for readability. Additional code comments are provided in the accompanying repository.

data are then interpolated using data from nearby stations, and a fix is also applied to any interpolated data where the maximum daily temperature is less than the daily minimum. The data from five of the selected stations are then written to Raven *.rvt
format. This workflow would be of substantial time and effort if performed manually or scripts for this task were adapted with each new application; in this use case, the entire workflow is performed with two main functions from the `weathercan` package and two from the `RavenR` package. The code required to accomplish this is provided in **Figure 2**.

The advantages of this workflow are 1) the ease of implementation, which can process any number of stations with only a few lines of R code; 2) the transparency and reproducibility of the *.rvt file generation, which is useful for both review of the
data and possible future corrections to all *.rvt files (e.g., extending the time series to incorporate more recent data). The code may be extended to any supplied set of stations and any meteorological variable that is recognized by Raven. The function also assumes standardized Raven parameter units for all meteorological variables (see reference tables in Appendix C of the Raven User's Manual (Craig, 2020)).

### 3.2.3  Observation data

Observation data, such as streamflow records, are generally not required to run hydrologic models; an exception to this may be for truncated model domains, where the model simulates a portion of the watershed and is supplemented by upstream measured flow data. However, observed time series are key to evaluating model performance (history matching) in both calibration and validation exercises and may also be used to enable data assimilation in forecasting applications.

Similar to the use of the `weathercan` R package for downloading Canadian meteorological data, the `tidyhydat` R
package may be used to download stream gauge data from Canadian stations maintained by the Water Survey of Canada (Albers, 2017). `RavenR` provides the `rvn_rvt_tidyhydat` function to process `tidyhydat` inputs directly by wrapping the `rvn_rvt_write` function, which can write any non-meteorological time series to *.rvt format. Possible types of time





series supported by Raven *.rvt files include reservoir inflows, irregular observations, observation weights, temporal reservoir operation rules, etc. The entire list of available formats can be found in the Raven User's Manual (Craig, 2020).

In this use case, the `tidyhydat` package is used to prepare *.rvt files of observed streamflow for 9 specified stations (consistent with the stations listed in Table 2 of Brown and Craig (2020)) used in the Liard model. The daily streamflow for these stations are downloaded using `tidyhydat` from 1985 to present day, and written to *.rvt format using the `rvn_rvt_tidyhydat` function (a wrapper for the `rvn_rvt_write` function). Raven will automatically exclude any missing values from the calculation of diagnostics, thus missing values in observation data generally do not need to be interpolated or infilled in the same

manner that meteorological forcing data needs to be processed. However, the user may still wish to be aware of and avoid large gaps in observation data that may bias the calculation of diagnostic metrics (e.g., consistent winter gaps or multi-year gaps).





```
obs_stns <- read.csv("observation_stations.csv")
tidyhydat::hy_daily_flows(station_number = obs_stns$stnID,
                          start_date = "1985-01-01") %>%
  rvn_rvt_tidyhydat(subIDs=obs_stns$subID)
```

**Figure 3.** Minimum code required for the use case described in **Section 3.2.3** of downloading and writing observed flow data into Raven *.rvt format using the `tidyhydat` and `RavenR` R packages. The pipe operator (`%>%`) from the `dplyr` package is used for readability. Observation station IDs and associated model subbasin IDs are provided in the 'observation_stations.csv' file for brevity. Additional code comments are provided in the accompanying repository.

The same `rvn_rvt_write` function may be used to write other *.rvt data types by adjusting the `rvt_type` parameter, which may be useful for writing the observation weights generated from the `rvn_gen_obsweights` function to exclude certain data periods from Raven diagnostics, as was done in the Liard model for winter periods with unreliable data records (Brown and Craig, 2020).

### 3.2.4 Model discretization file

The development of distributed and semi-distributed models require the discretization of a basin into homogeneous units representing hydrologically similar areas. This is typically completed through overlaying a number of spatial data sets which have a dominant effect on the hydrological response of the basin, such as land use, elevation, or soil information (step 2 in **Table 1**). In overlaying the spatial data sets, a large number of small computational units, insignificant to the model function, can be created. Since the model runtime is scaled with the number of HRUs, these small areas can increase computational and calibration run times and are not necessary to simulate the dominant hydrological response of the basin. The `RavenR` package offers a way to effectively eliminate small computational units using the `rvn_rvh_cleanhrus` function. This function may merge units based on a set area threshold, and can also merge similar HRUs based on similarity in HRU properties such as landcover, slope, elevation and aspect. HRUs which are significant to the model can be locked or protected. Locked HRUs cannot be removed from the model or increase in size and protected HRUs cannot be removed but may increase in size (to maintain the total watershed area) if other HRUs are removed. This is useful in cases where a point observation is available at a given location (snow survey data) or if the HRUs are part of a significant hydrological response (glaciers).

In this use case, the reduction in the number of model HRUs is demonstrated for a subset of the initial HRUs within subbasin 3 only (initially with 172 HRUs). HRUs that are of land use type GLACIER are locked HRUs (i.e. cannot be removed or change in size), and HRUs that are either WETLAND or WATER are protected (i.e. cannot be removed but can still increase in size if other HRUs are removed). This operation is applied using area thresholds of $0.5\%$ and $2.0\%$ of the subbasin area, resulting in 87 and 44 HRUs, respectively. The impact of this operation on land use distribution within the subbasin is summarized in **Figure 4**.



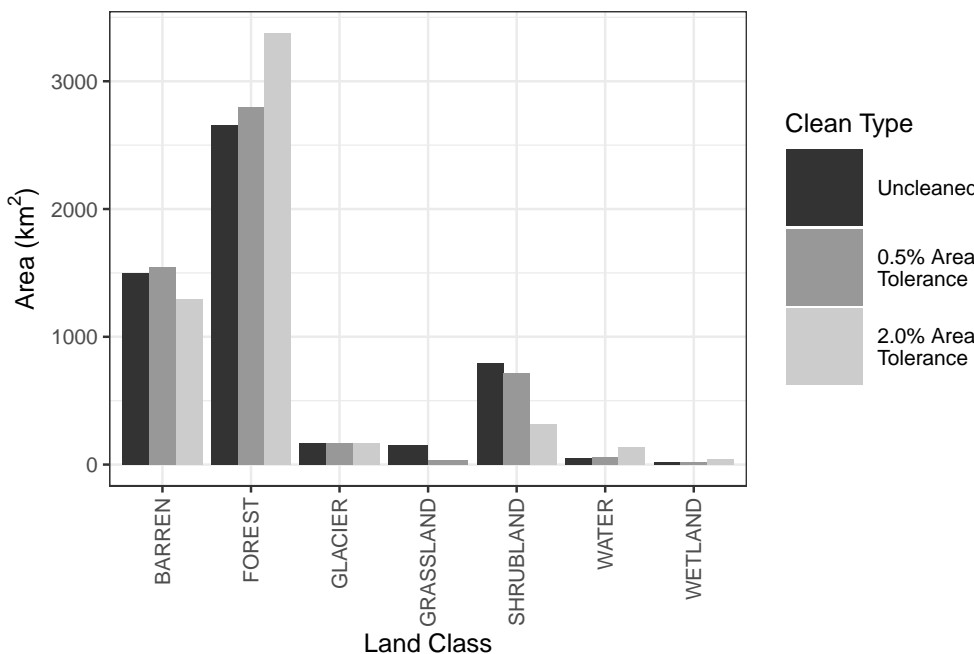

**Figure 4.** A barplot of total areas by land use for three sets of HRU configurations, including 1) prior to any 'cleaning', 2) following a 0.5% area threshold merging criteria with the `rvn_rvh_cleanhrus` function, and 3) following a 2.0% area threshold merging criteria. The GLACIER land use is locked, and the WETLAND and WATER land uses are protected. The example is done for a single subbasin in the watershed for demonstration purposes, and shows how the land use in the subbasin changes when the removal of subbasins below the area percentage threshold is performed using the `rvn_rvh_cleanrhus` function.

In the figure, the total area of all GLACIER land use type HRUs remains the same, as this land use type was locked. The WATER and WETLAND land use types increase slightly with each individual cleaning, relative to their proportion of the total area and the total area of removed HRUs based on the respective area threshold. It is noted that, were the WATER and WETLAND HRUs not protected (or locked), the area of these HRUs would be reduced with a $0.5\%$ threshold and removed entirely with a $2.0\%$ threshold. Literature has shown that hydrologic areas such as wetlands that are small in size can still have

a disproportionalety large effect on the hydrologic and biogeochemical response of the watershed (McLaughlin et al., 2014), thus retaining particular HRUs or HRU types may be critical in the cleaning of the HRUs. Finally, the plot shows how the other land use types change with these operations. The FOREST type increases in each case, suggesting that the proportion of small forested HRUs may be small, and that forested HRUs tend to be larger in size. The SHRUBLAND HRUs decrease in represented proportion in each case. This type of analysis could be repeated for other HRU properties (e.g. slope, aspect,

vegetation type, etc.). This analysis should be done in conjunction with the use of the `rvn_rvh_cleanrhus` function to ensure that the reduction in the number of HRUs does not unexpectedly alter the overall representation of HRUs within the model, and inhibit the ability of the model to capture the dominant hydrologic response.





## 3.3 Output file processing and analysis

A number of functions within `RavenR` are included to handle the reading of common Raven output files, such as the Hy-
drographs file (`rvn_hyd_read`), the WatershedStorage file (`rvn_watershed_read`), and other output files (forcings,
custom output, etc.). In addition, functions to analyze the output data with typical hydrologic checks and diagnostics are in-
cluded in the package. While these functions are built to work with the Raven-specific read functions they are otherwise not
specific to Raven, and may be used for any hydrologic model given a means of reading time series output is provided.

This section provides use cases where the realism of the Liard model is assessed, providing insight to the question, 'is the
model getting the right answers for the right reasons?' (Kirchner; Euser et al., 2013). These checks provide the modeller with
an understanding of the model dynamics and provide more confidence that the model is functioning as expected without model
compensation errors (step 7 in **Table 1**). This section also provides a demonstration of tools for evaluating model performance.

### 3.3.1 Evaluation of model realism

The flexibility of Raven in the generation of model outputs, including customized outputs that may be specified by the user,
can be leveraged to undertake rigorous checks of the hydrologic model. Tools have been built into the `RavenR` package to
capitalize on this feature, and facilitate a set of model reality checks. These checks can be considered semi-automatic, since
a script may be deployed to generate the figures but they still require interpretation by a modeller with an understanding of
both the natural system and the developed model. Here, the focus is on the realism of the model to ensure that it is providing
hydrologically plausible results; the actual performance of the model is discussed further in **Section 3.3.2**.

The checks that are applied to the Liard River basin in this use case include: 1) plotting the Budyko curve (Budyko, 1974)
for the annual average watershed indices, 2) plotting the annual regime curve with monthly averages, 3) examining the sta-
tionarity of moisture content in soil storage layers, and 4) plotting the model performance with respect to snowpack storage as
snow water equivalent (SWE). Additional checks supported (not demonstrated here) include: plotting the forcing functions to
understand how the inputs may be influencing the model results (i.e. wet and dry years, erroneous temperature readings, etc.),
checking the annual water balance, examining baseflow characteristics by comparing modelled and observed baseflow using
baseflow separation techniques, plotting annual hydrographs in an overlay (i.e. spaghetti plot), and checking the modelled
hydrographs and reservoir levels, if any reservoirs or lakes are included in the model.

The four plots associated with the stated checks performed in this example are provided in **Figure 5**.



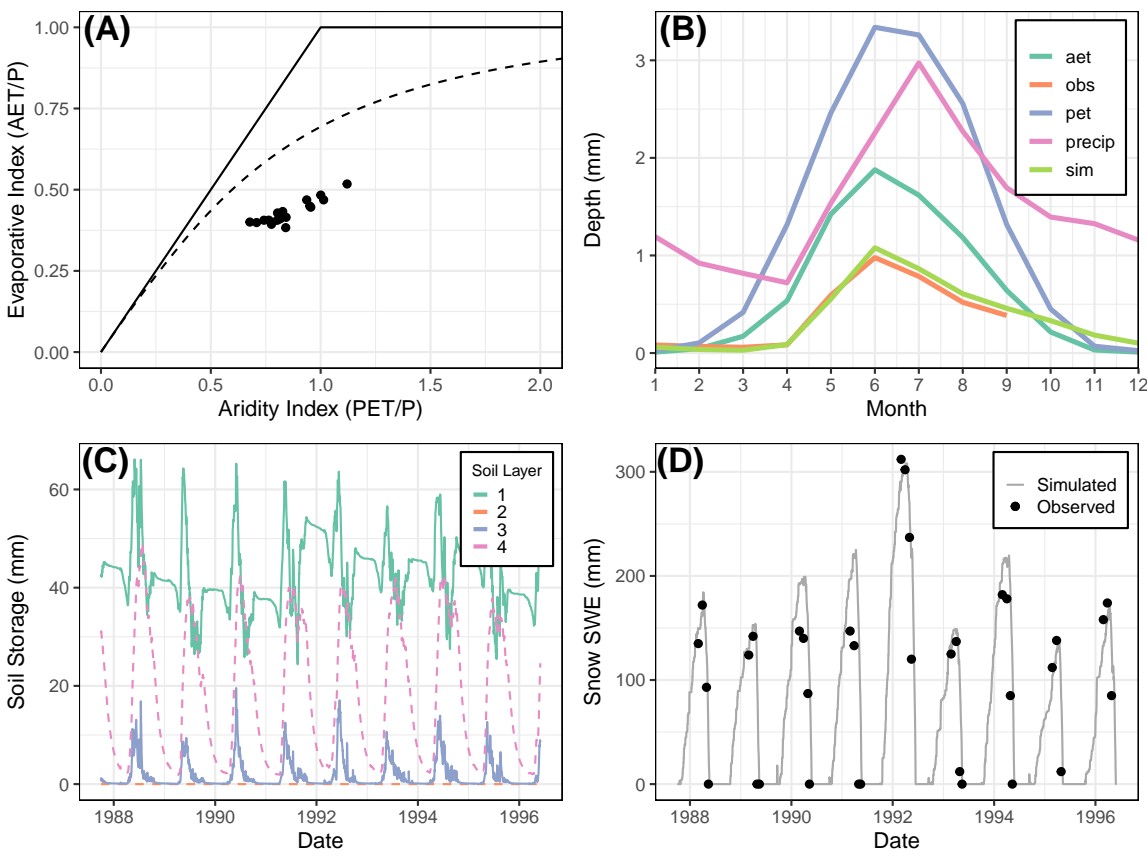

**Figure 5.** Multiple diagnostic plots generated from the `RavenR` package that may be useful in evaluating the realism of any hydrologic model; A) budyko curve with annual average data points for the watershed, B) a series of regime curves, C) soil storage time series showing the stationarity in long-term storage within soil layers and D) plots of observed and simulated snowpack measurements at Frances River. All data are generated from the Raven model averaged at the watershed scale unless otherwise indicated (i.e. snowpack SWE).





The Budyko plot in **Figure 5A** was generated using the `rvn_budyko_plot` function. The Budyko Curve (Budyko, 1974) shows the relationship that quantifies how mean annual precipitation is partitioned into discharge or evapotranspiration (ET). The Budyko pattern has been observed in multiple catchments around the world (Vereecken et al., 2015). The x-axis of the Budyko curve is the aridity index which represents the ratio between potential evapotranspiration and precipitation. The y-axis is the evaporative index and represents the ratio between actual evapotranspiration and precipitation. The curve is bound by two lines, shown in Figure 4a as the solid lines: the energy limiting line, which is when the evaporative index is equal to the aridity index, and the water limiting line which occurs when the actual evapotranspiration is equal to precipitation.

In a traditional application of the Budyko Curve it is expected that the plotted points would fall closer to the theoretical line shown in **Figure 5A** as the dashed line. However, certain traits of a basin, such as significant basin storage, can result in deviation from this line. The interpretation of the plot in Figure 4a could also indicate that actual evapotranspiration is being underestimated in the model. Alternatively, if there is significant inter-annual storage in the basin, perhaps due to the presence of wetlands, then the plot would be in line with reality. Ultimately, it is up to the modeller to decide whether this plotted behaviour is expected or if there is a possible misrepresentation of hydrologic processes in the model.

The regime curve can be used to examine the quantities and timing of some of the key model functions. For example, the **Figure 5B** shows that the PET in the Liard model (based on the Hargreaves 1985 calculation method, see Hargreaves and Samani (1985)) peaks at the same time as the AET in June, and maintains a similar pattern over the other months of the year on average. The simulated and observed flows are close in value, and both peak prior to the peak in precipitation. This aligns with the fact that peaks in the Liard River basin are typically freshet driven. Overall the plot provides a wealth of diagnostic information to the modeller, and a mismatch in elements of the regime curve, such as AET and PET or precipitation and flow that is not expected by the modeller would provide a point of investigation and validation.

The soil storage information can be retrieved from Raven in either the WatershedStorage.csv file (generated with the `:WriteMassBalanceFile` command), or with the custom output options for specific soil layers (e.g., `:CustomOutput DAILY AVERAGE SOIL[0] ENTIRE_WATERSHED`). Plots such as **Figure 5C** may be applied to any storage compartment in the model to verify the general assumption of long-term stationarity in storage within the hydrologic model. The stationarity assumption for a continuous simulation model is that over a long duration the soil storage should reach a quasi-equilibrium, oscillating around a steady mean. Therefore, a continuous simulation model which is continuously accumulating soil moisture during the simulation period may indicate that, for example, there is insufficient evapotranspiration or baseflow, resulting in the soil storage continuously increasing to compensate for this deficiency. A similar check for reservoir or lake storage could also be warranted. In the plot provided, all four soil storage units have a pattern that generally repeats annually, indicating the storage is stationary in the long term. In addition to verifying the stationarity assumption in storage and examining model compensation effects, **Figure 5C** may be used to better understand the soil dynamics in the model, and ensure that it is consistent with the understanding of the natural system. The pattern indicates that the top soil layer (layer 1) peaks in May-June (presumably with snowmelt) and rapidly depletes, while the fourth soil layer representing deep groundwater storage receives the infiltration at a more delayed rate and has a slower release from storage.





The snow plot provides a method to evaluate the snow balance representation in the model for a particular station. The simulated snow series is produced in Raven with the custom output command (e.g., `:CustomOutput DAILY AVERAGE`

`SNOW BY_HRU`), and is compared against the observed snow course measurements. The plot in **Figure 5D** was generated for the Frances River station. The snow plot provides a visual representation of the model's ability to represent the snow processes and compares it directly to observations. The model provides a reasonable representation of the snowpack SWE with no consistent bias in estimation. The same plots may be created wherever snow measurements are available, and provide a method for directly evaluating the snow routines in the model. In `RavenR`, the custom output snow is read into R

using the `rvn_custom_read` function, and irregular observations may be read using any native R read function (such as `read.table`) or the `rvn_rvt_read` function if they are provided as irregular observations in an *.rvt file. Raven may also directly calculate any model diagnostics for irregular snow observations (such as NSE) if they are provided to the model, which may be useful in building objective functions for automatic calibration of the watershed.

Although not included in the plots above, examination of the model input forcings is often an insightful step in diagnosing

potential model issues and realism. The forcing functions file that is outputted by Raven may be quickly read and visualized with `RavenR` (i.e. `rvn_forcings_read() %>% rvn_forcings_plot`) as a check. Issues such as errors in the temperature record or in the units of precipitation (mm/h instead of mm/d, for example) may be quickly determined with a visual check on the array of forcings that are both supplied to and determined by the Raven model.

### 3.3.2   Evaluation of model performance

The `RavenR` package provides a broad suite of tools for analyzing the results of any Raven hydrologic model, including many tools that can be considered model independent (step 7 in **Table 1**). For example, hydrograph plots, calculation of runoff coefficients, and flow duration curve plots are available within `RavenR` but may be computed for any time series of flows. The calculation of diagnostics, such as the commonly used Nash-Sutcliffe Efficiency (Nash and Sutcliffe, 1970) and Kling-Gupta Efficiency (Gupta et al., 2009) metrics, are not included in the `RavenR` package as they can be calculated directly within

Raven, and are also available comprehensively in existing packages such as `hydroGOF` (Mauricio Zambrano-Bigiarini, 2020).

In this use case, a number of diagnostic plots based on simulated and observed hydrographs are presented for the Liard River basin model. These diagnostic plots are computed at the outlet of the Liard River basin (at the outlet near Water Survey of Canada station 10ED1002), and are provided in **Figure 6**. These plots are provided for a portion of the simulation period (where the plot is time-based), and in practice these plots may be applied in both calibration and validation periods for comparison.



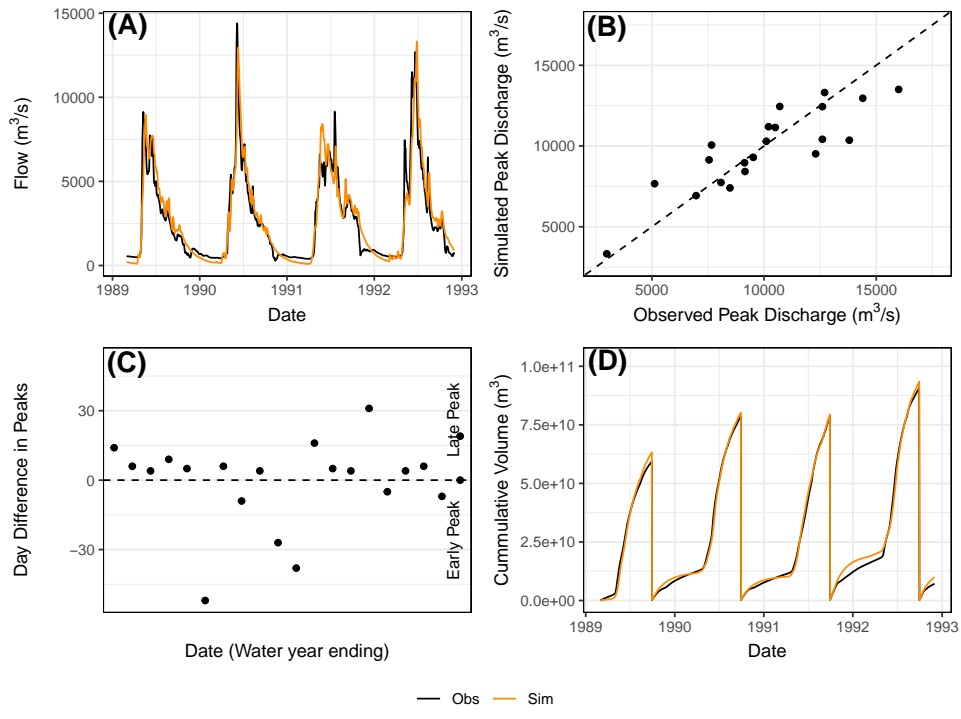

**Figure 6.** Multiple diagnostic plots generated from the `RavenR` package that may be useful in evaluating model performance; A) a hydrograph plot for a subset of the simulation period, B) a scatterplot of simulated and observed annual peak flows, C) plot of timing annual peak timing errors, and D) a plot of cumulative annual flow volumes in time. In plots A and D, the observed is plotted in black and the simulated in orange.





In **Figure 6A**, a simple hydrograph plot for a subset of the simulation period is provided. The hydrograph shows good agreement in the magnitude and timing of summer peaks for the years shown as well as the rising limb of the hydrograph, which was the focus of the calibration in the work of Brown and Craig (2020), with a tendency to overestimate the recession from the peak in June until late December/early January. The underestimation tends to continue until the next peak. The hydrograph is shown for a subset of a few years, rather than the entire period, as the full period can obscure the important

deviations of the simulated hydrograph from observations and mask deficiencies. Examining a smaller section of the plot (e.g. with the `prd` argument in many `RavenR` functions), such as only a few years of simulation or a particular event of interest, allows for a more critical examination of the nature of the model errors. A subset of a hydrograph can also be viewed dynamically as a dygraph in `RavenR` with the `rvn_hyd_dygraph` function, which is supported by the `dygraphs` package (Vanderkam et al., 2018).

The peak flow scatterplot (**Figure 6B**) is a scatterplot of the simulated and observed annual peak flows, calculated based on the October 1st water year and produced using the `rvn_annual_peak` function. This figure provides a visual assessment of the performance of modelled peak flow magnitudes, which can be seen as a function of the flow magnitudes themselves (i.e. does the model simulate peak flow magnitudes well at lower and higher flow values?). A model that is able to perfectly simulate peak flow magnitudes will have all points fall on the 1:1 line included in the plot. The plot also provides insight into

the nature of the peak error i.e. whether there is a systematic bias in over- or under-predicting peaks, or whether the errors are normally distributed. The plot provided shows that points are relatively close to the 1:1 line and there is no strong evidence of a systematic bias in flow magnitude prediction. However, since only 20 points are included on the plot, additional data may be required to produce conclusions that are statistically valid.

While (**Figure 6C**) captures the performance with respect to the magnitude of the flow peaks, the timing of peak flows is not

assessed. The plot in **Figure 6D** assesses the error in peak timing (rather than magnitude) with the `rvn_annual_peak_timing_error` function. A perfect model would have all points fall along the zero line, indicating that there is no error in the timing of predicted peaks. The results for the Liard simulation indicate that the model tends to predict peaks slightly later than the observed data, while some of the larger errors in timing tend to be in early peak prediction. In a forecasting framework, a data assimilation technique may reduce the timing (and magnitude) errors that are present in

the continuous simulation evaluated here. However, this tendency of the model may still be useful information for forecasters. The use of multiple functions in tandem within `RavenR` to examine both the peak magnitude and timing errors can be used to evaluate the model performance more comprehensively than a single function (see multiple `RavenR` functions named as `rvn_annual_*`).

Finally, **Figure 6D** provides a comparison of cumulative flow volumes between the simulated and observed model in time.

This plot is generated by the `rvn_cum_plot_flow` function. The plot shows clearly where deviations in the overall volume arise in time. For example, the plot for the Liard model shows that the December-March winter period of each year is a time of deviation in cumulative volumes, while the freshet-driven summer peak periods tend to match volume quite well overall. This is likely a result of the calibration procedure in (Brown and Craig, 2020), where ice affected flows in the winter were not considered in the calibration procedure due to high levels of uncertainty associated with the measurements. Additional




functions that examine the relative volumes of simulated and observed flows, but aggregate them rather than examining the differences in time, are the `rvn_monthly_vbias` and the `rvn_annual_volume` functions, which provide the monthly average volume differences and the annual volume differences in a scatterplot for each year, respectively. The volume is generally a useful diagnostic metric as it integrates the modelled hydrograph performance in time, and allows the modeller to identify periods of poor cumulative error or systematic errors (e.g. underestimating overall volume) that may be not clear or

obvious when only examining flows.

## 4   Conclusions

This paper presented the `RavenR` package, an R-based set of tools that is designed to support the development, use, and analysis of hydrologic models developed using Raven but can be readily adapted for any hydrologic modelling output. `RavenR` is a free, open-source software that is intended to support the wealth of options in a flexible modelling framework while

maintaining or improving the transparency and reproduciblity of the analyses undertaken by the modeller.

The tools within `RavenR` may be used in any stage of the typical modelling workflow. Although the tools are designed for use with Raven, the analysis and utility functions may be useful in conjunction with any hydrologic model that has similar requirements and workflows as Raven. The `RavenR` tools provide the means for preparing Raven input files, visualizing and processing input data, executing Raven, and generating a vast array of model checks and performance-related graphics from

the Raven output files. All functions in the package are supplemented by additional information and examples (consistent with CRAN requirements), and the package is further accompanied by the introductory documentation in the form of a vignette. This paper illustrates how the `RavenR` functions may be used in both academic and industrial projects, including generating model input *.rvt files, visualizing the model structure, and exploring and assessing the hydrologic model results. This includes aiding the modeller in building an understanding and trust of the constructed hydrologic model.

A set of `RavenR` use cases are presented for the Liard River basin, for which a Raven model has previously been built and thoroughly tested (Brown and Craig, 2020). The use cases present how a subset of tools may be used to generate input files for, or analyze the results of, the Raven model of the Liard river basin. The examples are bolstered by an interpretation of the functions and results, which may be useful in interpreting and building confidence in any hydrologic model. The accompanying data repository and code for this manuscript can fully recreate the figures and analyses presented in the use cases, demonstrating

best practices for reproducibility in hydrologic and scientific publications.

Due to the open-source nature of the Raven project, the code is transparent and accessible to users and is being continuously supplemented with new functionalities and improvements. Similarly, the `RavenR` package is open-source and is in active development. It is anticipated that the `RavenR` project will also continue to improve and expand its functionality in order to meet its goal of supporting Raven users from all backgrounds and experience levels while improving the reproducibility of

their work.

*Code and data availability.* The `RavenR` package is free and open-source software, and the version of the package (v.2.1.4) used to produce the results of this paper is archived on Zenodo (Chlumsky et al., 2021b). All R code and data used to generate the results and figures presented in this manuscript is also archived on Zenodo (Chlumsky et al., 2021a), and is also available on Github (https://github.com/rchlumsk/RavenR_manuscript_2021). The RavenR package is currently available as v2.1.4 on the Comprehensive R Archive
Network (CRAN) (https://cran.r-project.org/package=RavenR), and the development version of the package is also available on Github (https://github.com/rchlumsk/RavenR). The Raven hydrologic modelling framework is open-source and may be downloaded from http://raven.uwaterloo.ca/.

*Author contributions.* RC initiated the concept of the `RavenR` package. RC and JRC contributed the bulk of the package scripts, with contributions and development efforts from all authors. GB and JRC provided the Liard River model files. The use cases were prepared by
RC, LS, SL, SG, and GB. The article was prepared by RC with contributions from all authors.

*Competing interests.* The authors declare that they have no conflict of interest.

*Acknowledgements.* Mr. Chlumsky would like to acknowledge the financial support provided by the NSERC Canada Graduate Scholarship (CGSD3-558879-2021) and the Engineering Excellence Doctoral Fellowship provided at the University of Waterloo that helped make this work possible. The authors would like to thank all those who have contributed to the `RavenR` project since its inception, including Larry
(Haobo) Liu and Joel Trubilowicz.





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
