# Peer review of "RavenR v2.1.4: an open source R package to support flexible hydrologic modelling"

_Geoscientific Model Development, 2021_

## Author Comment (AC1)

We thank Mr. Astagneau for their comments, and for taking the time to review the submitted manuscript. Responses to each submitted comment are provided below, and we plan to make changes to the manuscript in response, as indicated in the author comments.

Original review comments are provided in *italics*, and author responses to the comments are in **bold** font in the sections below.

**0.1 Comments**

*This manuscript presents a new R package which aims at helping modellers in their use of the Raven hydrologic framework. Most of the package features consist in functions for data wrangling to feed Raven and functions for simulation analyses. Rationales behind the implementation of RavenR are presented. Examples of the RavenR functionalities are introduced using a formerly built perceptual model of the Liard river basin.*

*Several authors have advocated for the use of flexible structures for systematic testing of multiple working hypotheses in hydrological modelling. The use of such structures inherently results in higher complexity for modellers hence a challenge for reproducibility of methods and results. I think that any attempt at improving the use of these flexible structures is therefore relevant to the community of hydrological modellers. Furthermore, an extensive documentation is introduced to use the RavenR package, lots of interesting functionalities ranging from data preparation to simulation analysis are implemented and feedbacks between users and developers are encouraged to maintain and improve the package.*

*However, to be able to thoroughly evaluate the added value of using RavenR, I would have needed some experience with the Raven hydrologic framework. As it is not objectively possible in the time required to write a review, the following comments can only be seen as a way to improve the readability of the paper for non-Raven users and broaden the possible reach to the hydrological community.*

**0.2 General comments**

1. *Two similar flexible hydrological frameworks need to be cited in this work (either in the introduction or in Sect. 2): DECIPHeR (Coxon et al., 2019) and SuperflexPy (Dal Molin et al., 2020). A short description of the main differences between Raven/RavenR and these frameworks might further demonstrate the added value of using RavenR.*

   **Thank you for these additional citations to include. We agree that a short description of these differences would be well placed in the manuscript, and we plan to include this in the introduction or section 2 as suggested.**

2. *To improve understanding by new users of Raven (or even new hydrological modellers), I suggest adding a short description of the main choices that were made in the Raven hydrological framework and RavenR in terms of programming languages. The Raven hydrologic framework is coded in a compiled programming language, probably for computational speed and flexibility purposes. To improve its usability, the RavenR package was created. However, some hydrological models are coded in a compiled programming language and interfaced by R using packages (e.g. hydromad; Andrews and Guillaume, 2018). Why is the Raven workflow (in terms of programming languages) more suited for flexible modelling?*

   **This is worth emphasizing in the manuscript, and will help to partially address the discussion by RC2 as well. In short, the Raven framework is compiled in C++ for speed, and has many design features that put flexibility of the hydrologic model as the core consideration. The RavenR package is designed to improve the workflow with Raven, and uses the tools in R that are computationally slower to use but are perfectly suitable for analysis, and with the benefit of easier development. Due to the size, complexity, and rapid rate of development of the Raven source code, implementing and maintaining the Raven model within RavenR with the Rcpp library or a similar approach would be a massive undertaking with many technical challenges. Keeping the two separate allows for better code management and development of both software packages.**

3. *Section 3 is probably the most important section of this paper if we want to use the RavenR package and the Raven hydrologic framework. The steps of the hydrological workflow are presented in Table 1 and the*

*related R code and model files are provided to understand the functionalities of RavenR. However, I found some parts of this section a bit difficult to understand, especially since in the provided R script, the model run command line appears before input file processing.*

**The Rmarkdown file provided with the manuscript (RavenR_use_cases.Rmd) is intended to highlight and demonstrate certain functionalities of the RavenR package, rather than provide a complete sequential set of steps to develop all model input files and analyze all outputs. Thus, the model run call is made in the file prior to some of the sections to demonstrate input file processing and other tasks. This can be made more clear in the manuscript for section 3.**

4. *The authors state line 195 that step 4 and 5 will not be presented but it is not clear why. They are important steps of the hydrological workflow especially when performing uncertainty analyses. An explanation of why this is not relevant given the objectives of the paper is needed.*

   **The use cases focus on a subset of the available tools within RavenR, with an emphasis on how the package can be used to reduce the modeller's effort in working with Raven. A use case on running the model was not deemed to be required as it is a relatively straightforward command, and is shown in the Rmarkdown file provided in the repository - this can be mentioned in the manuscript.**

5. *Although it is probably relevant to introduce the notion of locked or protected HRUs in Sect. 3.2.4, hydrological modellers with less experience with Raven might need a simpler use case of model discretization first. If the authors want to keep this section as it is, I suggest adding a simpler example in the future vignettes of the package.*

   **This is a good comment to provide a simpler example for new users. We will consider updating the manuscript and use case file to show a first case where no locked or protected HRUs are provided, add this result to the plot, and update the discussion accordingly to highlight the difference. At a minimum, we will direct users to the function example in the package, which does not require the definition of locked or protected HRUs to run.**

6. *Sect 3.3 may be too long and its purpose not very clear since the evaluation of what the authors call "model realism" does not lead to questioning the hypotheses behind the Liard basin model. I think this section should be limited to a presentation of the possible analyses of model simulation enabled by RavenR. Possible cuts: l 376 to l 381; l 383 to "Overall" l 386; from "A similar check" l 396 to l 402; from "The model" l 407 to "bias in estimation" l 408; from "The hydrograph" l 430 to "peak" l 433; from "The plot" l 446 to l 448; from "The results" l 452 to l 453; from "The plot shows" l 460 to "measurements" l 464.*

   **We thank you for this recommendation, and agree that section 3.3 should be reduced to be more succinct, and perhaps reduce the number of figures and related explanations in the current version of the manuscript. We will take into account these recommendations in producing a revised version.**

7. *Overall, I think that the R script provided to understand Sect. 3 could become a vignette but for a very simple use case that would include parameter estimation procedures and questioning of modelling hypotheses. Building a simple model from data preparation to output analysis using a catchment from the Camel dataset (Addor et al., 2017) would allow very different modellers to use the Raven hydrologic framework.*

   **The RavenR software is not necessarily intended to build every required or desired input file from scratch in R, but to provide tools to make this process more efficient for users. The manuscript is to demonstrate some of these useful tools. We do not think this manuscript is the best place for a complete tutorial on building model files from scratch, however, we can point to the existing RavenR package vignette and the Raven User's Manual as useful resources in this respect. We also agree that an accompanying vignette for the package that walks a user through these steps would be a very useful addition, and we plan to include this in future versions of the software.**

**0.3 Minor comments**

1. *I think that lines 60 to 70 could be moved just after line 44 for better links between the paragraphs of the introduction.*

**Thank you for this suggestion, we agree and will update this in the revised manuscript.**

2. *Please add the references of Python, R and C++.*

   **This will be updated in the revised manuscript.**

3. *Line 128, "3) running raven" should be moved before "2) reading output files".*

   **Thank you, this will be updated in the revised manuscript.**

4. *Line 349/350: please remove "providing...for the right reasons? (Kirchner; Euser et al., 2013)", as it is not the place to provide insights into a scientific question that was not presented in the introduction.*

   **This will be updated in the revised manuscript, and done in tandem with providing better definition for model realism and reality checks.**

5. *Please define "model realism" and "reality checks" in Sect. 3.3.1, as they are vague concepts, especially when no other data than streamflow are available for model validation.*

   **Clarification on the use of these terms will be provided in the manuscript, along with a note about limiting these checks to the available data (i.e. streamflow observations).**

6. *Line 365: I do not think that the term "observed baseflow" can be used to refer to the results of baseflow separation techniques that rely only on streamflow time series.*

   **Thank you for this point, the term for "observed baseflow" will be updated in the revised manuscript to ensure this distinction is clear.**

7. *Lines 414 to 418 should not appear in Sect 3.3.*

   **These lines will be removed from section 3.3, and consider instead making a note in section 3.2.2 about how forcing data may be read in from Raven output as part of a workflow.**

8. *Line 449: "Figure C" should be "Figure B".*

   **Thank you, this will be reflected in the revised manuscript.**

9. *Line 550: "Figure D" should be "Figure C".*

   **Thank you, this will be reflected in the revised manuscript.**

**0.4 Technical comments**

*I noticed a few typos. As I am a non-native English speaker, the following comments might not be relevant.*

1. *L1: "advances...have enhanced" instead of "has enhanced".*

   **Thank you for this, the correction is valid and will be updated.**

2. *References such as "(e.g. GR4J (Perrin et al., 2003))" should appear as "(e.g. GR4J; Perrin et al., 2003)". The latex command for this is:*
   *citep[e.g. GR4J;][]citationkey.*

   **Thank you, this will be reflected in the revised manuscript.**

3. *Line 312: "The development...requires" instead of "require".*

   **Thank you for this, the correction is valid and will be updated.**

**0.5 Comments specific to the R package documentation**

1. *From my understanding, the pipe operator is not mandatory to run the Raven package and is only used here for better readability. However, some R users are not familiar with the dplyr syntax. Although this is mentioned in the title of Figure 2 of the article, I would recommend adding this information in the package documentation (if not done already, I might have missed it).*

   **Thank you, this is correct and will be noted in the package documentation (including the package vignette).**

2. *For some functions (e.g. rvn_annual_peak), the units of the related arguments are mentioned in the detail section. It is always easier for users to find the required unit beside the related argument. I would suggest doing so in future versions of the package.*

   **We agree, and will be sure to include units where possible in future updates to the package.**

3. *I noticed that for some functions, time series must be provided at a daily time step. I thought that the Raven hydrologic framework could run at multiple time steps. Again, I might have missed the explanation at some point. If not, I would suggest adding a warning somewhere to use the time step required by RavenR/Raven.*

   **Raven can indeed be run at any time step that is less than or equal to one day. The functions are generally setup to run for any time step but we will be sure to include warnings for those that require a daily time step in future versions of the package. Thank you for this note.**

---

## Author Response (AR1)

We thank the reviewers for their comments. In addressing these comments, we have made the following key changes to the manuscript:

1. The manuscript has been streamlined in a number of sections, including in Section 1 (Introduction) and Section 3 (Use Cases).

2. A discussion on the interaction between Raven and RavenR software, including the choice of keeping these decoupled, has been added in Section 2.2.1 (RavenR Overview).

3. The analysis in Section 3.2.4 (Model discretization file) has been updated, including the plot, discussion, and repository script to reflect comments of Reviewer 1.

We think that the manuscript now provides a more streamlined read, and includes some additional clarifying details to address potential questions about the software that were raised by reviewers.

We answer specific reviewer comments in the following manner. Comments are in *italic*; answers are in **bold** font. **Red line numbers** refer to the plain new version of the manuscript, and **purple line numbers** to its track-changed version.

We need to mention that the provided LaTeX template `copernicus.cls` shows some bugs in the citations, numbering and referencing of sections, equations, figures, captions, and tables when the commands `\add{}`, `\remove{}`, `\change{}`, etc. are used. In the track changes version of the document, some added portions are shown in this colour, and text to be removed is shown in this color, in addition to the default settings for the `\add{}`, `\remove{}`, and `\change{}` commands.

**1 Author Responses to Reviewer 1**

We thank Mr. Astagneau for their comments, and for taking the time to review the submitted manuscript. Responses to each submitted comment are provided below, and we plan to make changes to the manuscript in response, as indicated in the author comments.

Original review comments are provided in *italics*, and author responses to the comments are in **bold** font in the sections below.

**1.1 Comments**

*This manuscript presents a new R package which aims at helping modellers in their use of the Raven hydrologic framework. Most of the package features consist in functions for data wrangling to feed Raven and functions for simulation analyses. Rationales behind the implementation of RavenR are presented. Examples of the RavenR functionalities are introduced using a formerly built perceptual model of the Liard river basin.*

*Several authors have advocated for the use of flexible structures for systematic testing of multiple working hypotheses in hydrological modelling. The use of such structures inherently results in higher complexity for modellers hence a challenge for reproducibility of methods and results. I think that any attempt at improving the use of these flexible structures is therefore relevant to the community of hydrological modellers. Furthermore, an extensive documentation is introduced to use the RavenR package, lots of interesting functionalities ranging from data preparation to simulation analysis are implemented and feedbacks between users and developers are encouraged to maintain and improve the package.*

*However, to be able to thoroughly evaluate the added value of using RavenR, I would have needed some experience with the Raven hydrologic framework. As it is not objectively possible in the time required to write a review, the following comments can only be seen as a way to improve the readability of the paper for non-Raven users and broaden the possible reach to the hydrological community.*

**1.2 General comments**

1. *Two similar flexible hydrological frameworks need to be cited in this work (either in the introduction or in Sect. 2): DECIPHeR (Coxon et al., 2019) and SuperflexPy (Dal Molin et al., 2020). A short description*

*of the main differences between Raven/RavenR and these frameworks might further demonstrate the added value of using RavenR.*

**Thank you for these additional citations to include. We agree that a short description of these differences would be well placed in the manuscript. We have added reference to the DECIPHeR framework in the Introduction (Line 24), and mentioned SuperflexPy in additional clarification about the package in Section 2 (specifically, Section 2.2.1., Line 110), as suggested.**

2. *To improve understanding by new users of Raven (or even new hydrological modellers), I suggest adding a short description of the main choices that were made in the Raven hydrological framework and RavenR in terms of programming languages. The Raven hydrologic framework is coded in a compiled programming language, probably for computational speed and flexibility purposes. To improve its usability, the RavenR package was created. However, some hydrological models are coded in a compiled programming language and interfaced by R using packages (e.g. hydromad; Andrews and Guillaume, 2018). Why is the Raven workflow (in terms of programming languages) more suited for flexible modelling?*

    **This is worth emphasizing in the manuscript, and will help to partially address the discussion by RC2 as well. In short, the Raven framework is compiled in C++ for speed, and has many design features that put flexibility of the hydrologic model as the core consideration. The RavenR package is designed to improve the workflow with Raven, and uses the tools in R that are computationally slower to use but are perfectly suitable for analysis, and with the benefit of easier development. Due to the size, complexity, and rapid rate of development of the Raven source code, implementing and maintaining the Raven model within RavenR with the Rcpp library or a similar approach would be a massive undertaking with many technical challenges. Keeping the two separate allows for better code management and development of both software packages. This discussion is emphasized in Section 2.2.1 of the revised manuscript.**

3. *Section 3 is probably the most important section of this paper if we want to use the RavenR package and the Raven hydrologic framework. The steps of the hydrological workflow are presented in Table 1 and the related R code and model files are provided to understand the functionalities of RavenR. However, I found some parts of this section a bit difficult to understand, especially since in the provided R script, the model run command line appears before input file processing.*

    **The Rmarkdown file provided with the manuscript (RavenR_use_cases.Rmd) is intended to highlight and demonstrate certain functionalities of the RavenR package, rather than provide a complete sequential set of steps to develop all model input files and analyze all outputs. Thus, the model run call is made in the file prior to some of the sections to demonstrate input file processing and other tasks.**

4. *The authors state line 195 that step 4 and 5 will not be presented but it is not clear why. They are important steps of the hydrological workflow especially when performing uncertainty analyses. An explanation of why this is not relevant given the objectives of the paper is needed.*

    **The use cases focus on a subset of the available tools within RavenR, with an emphasis on how the package can be used to reduce the modeller's effort in working with Raven. A use case on running the model was not deemed to be required as it is a relatively straightforward command, and is shown in the Rmarkdown file provided in the repository. Reference to this file has been added on Line 185.**

5. *Although it is probably relevant to introduce the notion of locked or protected HRUs in Sect. 3.2.4, hydrological modellers with less experience with Raven might need a simpler use case of model discretization first. If the authors want to keep this section as it is, I suggest adding a simpler example in the future vignettes of the package.*

    **This is a good comment to provide a simpler example for new users. We have updated the manuscript (Section 3.2.4) and use case file to show a first case where no locked or protected HRUs are provided, added this result to the plot, and updated the discussion accordingly to highlight the difference. The repository for the manuscript has been updated accordingly.**

6. *Sect 3.3 may be too long and its purpose not very clear since the evaluation of what the authors call "model realism" does not lead to questioning the hypotheses behind the Liard basin model. I think this*

*section should be limited to a presentation of the possible analyses of model simulation enabled by RavenR. Possible cuts: l 376 to l 381; l 383 to "Overall" l 386; from "A similar check" l 396 to l 402; from "The model" l 407 to "bias in estimation" l 408; from "The hydrograph" l 430 to "peak" l 433; from "The plot" l 446 to l 448; from "The results" l 452 to l 453; from "The plot shows" l 460 to "measurements" l 464.*

**We thank you for this recommendation, and agree that section 3.3 should be reduced to be more succinct. We have made some substantial cuts to Section 3.3 to reduce the length of the section (e.g. Section 3.3.1 Lines 397-399, 403-407, 423-427, 434-444, including many of the recommendations suggested.**

7. *Overall, I think that the R script provided to understand Sect. 3 could become a vignette but for a very simple use case that would include parameter estimation procedures and questioning of modelling hypotheses. Building a simple model from data preparation to output analysis using a catchment from the Camel dataset (Addor et al., 2017) would allow very different modellers to use the Raven hydrologic framework.*

**The RavenR software is not necessarily intended to build every required or desired input file from scratch in R, but to provide tools to make this process more efficient for users. The manuscript is to demonstrate some of these useful tools. We do not think this manuscript is the best place for a complete tutorial on building model files from scratch, however, we can point to the existing RavenR package vignette and the Raven User's Manual as useful resources in this respect. We also agree that an accompanying vignette for the package that walks a user through these steps would be a very useful addition, and we plan to include this in future versions of the software.**

**1.3 Minor comments**

1. *I think that lines 60 to 70 could be moved just after line 44 for better links between the paragraphs of the introduction.*

**Thank you for this suggestion, we agree and has been updated in the revised manuscript Lines 43-53.**

2. *Please add the references of Python, R and C++.*

**This has been updated in the revised manuscript (Lines 36 & 82).**

3. *Line 128, "3) running raven" should be moved before "2) reading output files".*

**Thank you, has been updated in the revised manuscript (Line 118).**

4. *Line 349/350: please remove "providing...for the right reasons? (Kirchner; Euser et al., 2013)", as it is not the place to provide insights into a scientific question that was not presented in the introduction.*

**This has been updated in the revised manuscript (Lines 369-370).**

5. *Please define "model realism" and "reality checks" in Sect. 3.3.1, as they are vague concepts, especially when no other data than streamflow are available for model validation.*

**We have provided a definition in the opening of Section 3.3 that draws on the definition in Euser et al. [2013] to clarify our meaning (Line 330), and removed reference to "reality checks" in order to avoid confusion (e.g., Line 9).**

6. *Line 365: I do not think that the term "observed baseflow" can be used to refer to the results of baseflow separation techniques that rely only on streamflow time series.*

**Thank you for this point, the term for ""observed baseflow" has been updated to 'estimated baseflow" in the revised manuscript (Line 339).**

7. *Lines 414 to 418 should not appear in Sect 3.3.*

**These lines have been removed from section 3.3, Lines 440-444.**

8. *Line 449: "Figure C" should be "Figure B".*

**Thank you, has been reflected in the revised manuscript (Line 475).**

9. *Line 550: "Figure D" should be "Figure C".*

**Thank you, this has been reflected in the revised manuscript (Line 476).**

**1.4 Technical comments**

*I noticed a few typos. As I am a non-native English speaker, the following comments might not be relevant.*

1. *L1: "advances. . . have enhanced" instead of "has enhanced".*

   **Thank you for this, the correction is valid and has been updated (Line 1).**

2. *References such as "(e.g. GR4J (Perrin et al., 2003))" should appear as "(e.g. GR4J; Perrin et al., 2003)".
   The latex command for this is:
   citep[e.g. GR4J;][]citationkey.*

   **Thank you, this has been reflected in the revised manuscript, e.g., Line 18.**

3. *Line 312: "The development. . . requires" instead of "require".*

   **Thank you for this, the correction is valid and has been updated ((Line 330)).**

**1.5 Comments specific to the R package documentation**

1. *From my understanding, the pipe operator is not mandatory to run the Raven package and is only used here for better readability. However, some R users are not familiar with the dplyr syntax. Although this is mentioned in the title of Figure 2 of the article, I would recommend adding this information in the package documentation (if not done already, I might have missed it).*

   **Thank you, this is correct and will be noted in the package documentation (including the package vignette).**

2. *For some functions (e.g. rvn_annual_peak), the units of the related arguments are mentioned in the detail section. It is always easier for users to find the required unit beside the related argument. I would suggest doing so in future versions of the package.*

   **We agree, and will be sure to include units where possible in future updates to the package.**

3. *I noticed that for some functions, time series must be provided at a daily time step. I thought that the Raven hydrologic framework could run at multiple time steps. Again, I might have missed the explanation at some point. If not, I would suggest adding a warning somewhere to use the time step required by RavenR/Raven.*

   **Raven can indeed be run at any time step that is less than or equal to one day. The functions are generally setup to run for any time step but we will be sure to include warnings for those that require a daily time step in future versions of the package. Thank you for this note.**

**2 Author Responses to Reviewer 2**

We thank the anonymous reviewer for their comments, and for taking the time to review the submitted manuscript. Responses to each submitted comment are provided below, and we plan to make changes to the manuscript in response, as indicated in the author comments.

Original review comments are provided in *italics*, and author responses to the comments are in **bold** font in the sections below.

**2.1 General comments**

*This manuscript provides a description of a set of R functions to process in- and output files and the contained data, for the purpose of running the hydrological modelling software Raven, which itself is available as a C++ executable.*

*The manuscript is generally well-presented and well written, and I have very few specific comments. However, I make the following more general observations:*

1. *I am actually not sure whether the manuscript fits any of the designated manuscript type. I suppose that it is classified as a development paper because of the ample references to reproducibility in the manuscript. However, the original model is available as an open source code as well and therefore perfectly reproducible (at least in the sense that it is described on the about GMD web page). So at most, it is an enhancement of the usability of a specific model rather than its reproducibility.*

   **The manuscript type was changed from its original submission to a development paper by the handling editor, so we respectfully suggest that this is the best categorization of the manuscript type, based on the discussion of technical aspects of running models and reproducibility of results.**

2. *The other reviewer has made some very useful comments on the presentation, with which I fully agree. Overall, I think that the manuscript is too wordy and can be reduced substantially. Specifically, the authors seem to be at pains to convince the reader about the importance of open source software, accessibility, and good practices in model development. I don't think that the GMD readership needs such advocacy. It distracts from the core message and makes the manuscript unnecessarily long and somewhat tedious to read. (For example, the section L.137 - 146 is quite trivial and may be deleted entirely, but also many other sections can be streamlined).*

   **Some sections of the manuscript have been reduced in the revised version based on the comments of both reviewers, including adjustments to L. 137-146 as suggested (e.g., Lines 40-43, 91-94, 112-114, 137-144).**

3. *The technical implementation of the package is quite straightforward, and does not make optimal use of advanced functionality of R.*

   **Author responses are provided to the specific comments in the following section on Technical Comments.**

**2.2    Technical Comments**

*Specifically:*

1. *The fact that the model needs to be run separately is not very elegant. It would be ideal if the Raven model itself is distributed with the package as a dynamic library, and can be loaded as such by the R process. This would avoid the need for separate installation of the model, as well as the slightly clunky way that the executable is called by the rvn_run() function. It would also help with the next point.*

   **This point is brought up by reviewer 1 as well, and although it may be theoretically possible to include the entire Raven project in the RavenR distribution and compile it through the Rcpp library in R (or similar approach), there are at least two technical reasons why this is not ideal aside from the fact that this is not the stated purpose of the package. First, the size and complexity of the existing Raven code would necessitate a massive undertaking to import the project and ensure it can compile in R. Second, the Raven framework is used by a variety of users, and this could create technical issues for users that have their own (non-RavenR) workflows and forecasting environments with Raven, such as those running Raven with direct shell interaction on clusters or with python tools (e.g., using the RavenPy API). Keeping a copy of the code in RavenR as a non-master version may be possible, but would lead to code management duplication and additional overhead. Installation of Raven is trivial, as it simply requires the presence of the self-contained executable file. The functions to run the executable operate similar to other well-known hydrologic modeling packages like flopy [Bakker et al., 2016]. However, improved integration with RavenR and other scripting languages is a worthwhile future goal for Raven that is under development. A bit of text has been added to Section 2.2.1, Lines 109-116,** to discuss some of this reasoning for the separation in softwares.

2. *The fact that the scripts writes the input files to disk, which are then subsequently read by the executable (and vice versa for the output files) is inelegant at least, and probably also inefficient as well. If the model itself were implemented as a dynamic library then the in- and output data could be passed in memory to the model, which would greatly enhance performance in use cases such as monte carlo simulation.*

The authors agree that this would certainly be more efficient for projects requiring many model runs, and that this is a computational limitation of the current software implementation. However, the effort to convert Raven into a dynamic library is significant and well outside of the scope of this paper. For such work, other tools and scripts with less overhead may be preferable. Many of the use cases discussed in the paper relate to preparing input files and analysing final output files, which would not suffer the same limitations as described above. We have also used RavenR successfully to process the outputs from tens of thousands of calibration experiments, and found the computational demand of this read-write process to be acceptable. Raven does include comprehensive features for controlling the frequency and extent of output files, which can greatly help in keeping voluminous sensitivity/uncertainty exploration run times to a minimum.

3. *The package makes relative limited use of the object oriented nature of R. It does use relevant classes such as xts and lubridate, but does not define any classes itself. This results in a very long list of functions, essentially one function for every step in the analysis. It would be much more elegant (and efficient) to define a set of classes (e.g., one for each in- and output file, by extending classes such as xts) and then use method dispatch to read and write them, as well as any other standard processing such as aggregation. This would reduce the need for a long list of different functions to a few read() and write() commands, and allow for method dispatch on existing xts functions.*

   We agree that this would be a more efficient and elegant way to organize the package and make use of the more advanced aspects in R as mentioned, but by no means necessary to provide access to useful functionality. This will be strongly considered in a major version update for future package revisions.

*Lastly, while the examples in the manuscript are generally easily reproducible, some of the examples in the online documentation are not, for example because they include idiosyncratic path statements. i strongly recommend the authors to read through R guidelines such as the ones below, and cross-check that all the code adheres to these good practices:*

*https://www.tidyverse.org/blog/2017/12/workflow-vs-script/*

*https://www.carlboettiger.info/2013/06/13/what-I-look-for-in-software-papers.html*

The RavenR package is compliant with the rather rigorous standards of CRAN, and the examples are being continually updated for clarity. The authors will perform a search for idiosyncratic path statements and revise them based on the standards linked above in future versions of the software. We thank the reviewer for directing us to these resources.

**2.3 Conclusion**

*To conclude, I believe that this is certainly a useful piece of software, however for me the manuscript reads too much like a manual instead of a scientific paper, even of the type that GMD aims at. I think that there is scope for streamlining, and ideally going a bit beyond simply presenting a wrapper, towards exploring how even something as simple as a wrapper can incorporate state-of-the-art software design concepts. This does not mean that the software needs to be entirely implemented according to the recommendations above. But some attempt, or at least a discussion as to why this may be scentifically non-trivial, would lift the scientific value of the manuscript in my opinion.*

We have certainly streamlined the revised version of the manuscript based on the recommendations of the reviewers, and included notes Sections 2.1 and 2.2.1 to make clear the interplay between Raven and RavenR. We note here that the focus of this manuscript (and our development of RavenR) is not on incorporating state-of-the-art software design, but rather providing a software contribution which benefits the state-of-the-practice.

**References**

M. Bakker, V. Post, C. D. Langevin, J. D. Hughes, J. T. White, J. J. Starn, and M. N. Fienen. Scripting modflow model development using python and flopy. Groundwater, 54(5):733–739, 2016. doi: https://doi. org/10.1111/gwat.12413. URL `https://ngwa.onlinelibrary.wiley.com/doi/abs/10.1111/gwat.12413`.

T. Euser, H. C. Winsemius, M. Hrachowitz, F. Fenicia, S. Uhlenbrook, and H. H. G. Savenije. A framework to assess the realism of model structures using hydrological signatures. Hydrology and Earth System Sciences, 17(5):1893–1912, 2013. doi: 10.5194/hess-17-1893-2013. URL `https://hess.copernicus.org/articles/17/1893/2013/`.

---

## Author Response (AR2)

We thank the editor and reviewers for their additional comments. In addressing these comments, we have made the following key changes to the manuscript:

1. The manuscript has been further streamlined in the Introduction, reducing the discussion on reproducibility.

2. Section 2.2.1 has been updated as per the technical comments of Reviewer 2 surrounding the compilation of C++ code in R.

3. Minor text edits in other sections as per reviewer comments, detailed in this letter.

We think that the manuscript has now addressed the concerns of reviewers, and provides a more streamlined read, particularly in the Introduction.

We answer specific reviewer comments in the following manner. Reviewer comments are in *italic*; answers are in **bold** font. **Red line numbers** refer to the plain new resubmitted version of the manuscript, and **purple line numbers** to its resubmitted track-changed version.

We need to mention that the provided LaTeX template `copernicus.cls` shows some bugs in the citations, numbering and referencing of sections, equations, figures, captions, and tables when the commands `\add{}`, `\remove{}`, `\change{}`, etc. are used. In the track changes version of the document, some added portions are shown in this colour, and text to be removed is shown in this color, in addition to the default settings for the `\add{}`, `\remove{}`, and `\change{}` commands.

**1 Author Responses to Reviewer 1**

We thank Mr. Astagneau for their comments, and for taking the time to review the re-submitted manuscript.

Since no further comments were provided by Mr. Astagneau that require addressing, no additional responses are provided here for Reviewer 1.

**2 Author Responses to Reviewer 2**

We thank the anonymous reviewer for their additional comments, and for taking the time to review the re-submitted manuscript. Responses to each submitted comment are provided below.

Reviewer comments are provided in *italics*, and author responses to the comments are in **bold** font.

**2.1 Technical Comments**

*Line numbers refer to the version with tracked changes.*

1. *l49: shapefiles are a type of spatial data, so they don't need to be mentioned separately.*

   **This line has now been removed entirely as part of efforts to streamline the Introduction.**

2. *l71 - 81: This section is a repetition of l45 - 55. Comparison with the manuscript without tracked changes learns that text in red has been removed?*

   **That is correct, this section in red from the previous submission has been removed already.**

3. *l132: "maintaining the computational speed": why would this be? As mentioned in my previous comments, one can include C++ code in an R package and achieve exactly the same speed as a separate binary (while avoiding the overhead of reading and writing in- and output files). Perhaps the authors refer here to a model fully implemented in R, but that point is irrelevant here because of the excellent interfaces of R with compiled code. So I do not think that anyone would really consider translating C++ code into R just to include it in a package.*

There would likely still be some overhead in R, though we agree that compiling the Raven code in R would generally maintain its speed, and we are not suggesting that the entire library would be rewritten in R. However, compiling Raven in R would be a massive technical undertaking, let alone ensuring that Raven continues to compile in R with each update of R libraries and frequent changes in Raven code. We have adjusted lines 112 to 117 in acknowledgement of the reviewer comments, and simply stated that this separation allows for parallel development without these technical compilation and code development challenges, which also shortens this section.

4. *l134: this point does not really hold. Even if a compiled version of the model is included as a dynamic library in the package, then it is still perfectly feasible to install the model separately for non-R workflows. The R library will not be visible outside the R environment and so will not interfere with any other installations.*

   **We assert that building non-R workflows for a library contained in R would be less likely to occur, and having a pure C++ library is a cleaner way to approach this portability issue without simply maintaining two versions of the same code project (i.e. one wrapped in an R project and one separate). However, we have adjusted this section in light of the reviewer comments.**

5. *l634: "Mauricio Zambrano-Bigiarini" -¿ Zambrano-Bigiarini, M.*

   **This has been updated in our citations and bibliography.**

**3 Author Responses to Reviewer 3**

We thank the anonymous reviewer for their comments, including their congratulatory note, and for taking the time to review the re-submitted manuscript. Responses to each submitted comment are provided below.

Reviewer comments are provided in *italics*, and author responses to the comments are in **bold** font.

**3.1 General comments**

*The manuscript provides a detailed overview about the RavenR model setup and evaluation package. The authors document and demonstrate the application of this tool within the RavenR framework for the example of a specific river basin.*

*First of all, congrats to the authors for putting together such an extensive tool to facilitate the setup, use and evaluation of hydrological models generated within the Raven framework. I think this paper is in a good state for publication. While the tool itself is rather specific to one modeling framework, the publication of such a tool is a good blueprint for other modeling groups to develop and improve similar scripts on their own and, thus, earns its place in this journal. I only have some minor comments that the authors may use to improve the manuscript, but the publication should not be conditional to that:*

1. *it might be sensible to more clearly define the intended user group of the tool right from the start. The abstract and introduction mention the potential application of RavenR for hydrological models outside the Raven framework. While this is technically true at least for such evaluation packages that only require a single time series, these evaluations are usually also found in different evaluation packages or are probably already included in the workflows of other models. Here, RavenR does not appear to be worth the effort to write output conversions scripts for other models. However, RivenR really shines as a comprehensive support software for the Raven framework. As a hydrological modeler who neither uses R nor Raven or any of the hydrological models mentioned in the manuscript, the paper is still an interesting and inspiring lecture. However, I don't feel I can profit from this tool at all with an reasonable effort. In order to manage expectations, I would mention any general application possibility only in the conclusion and otherwise target the Raven user group more directly.*

   **This is fair point, and we have updated the abstract (line 7) to tone down the applicability for non-Raven users, and ensured that The Introduction in lines 68 to 75 and Section 2.2 refer to the use of RavenR specifically to support hydrologic models built with Raven, rather than more general use.**

2. *section 3.2.2 appears a bit too optimistic to me. True, as long as a tool like weathercane is available RivenR can utilize its standardized interface and data format. However, as soon as user target river basins in different regions, such tools will either be not available at all or use different data formats which will require a considerable effort from user side to adapt it for working with RavenR. Such limitations should be mentioned clearly.*

   **We have added lines 250 to 252 in Section 3.2.2 to address this point, which is certainly worth mentioning in this section to avoid an overly optimistic perspective on data handling outside of the defaults.**

3. *Even after the revision, the manuscript feels quite long in parts of the introduction. While I very much sympathize with the authors call for transparent and open(-source) science, this statement feels a bit out of place in a journal like this, as I would assume almost all readers already share this view. One the other hand, it cannot hurt to emphasize it once again.*

   **This sentiment was also echoed by other reviewers and the editor, thus we have substantially trimmed down this discussion in the introduction (see lines 38 to 67) to streamline the document. Note that this text appears as red text in the track changes document, rather than strikethrough font, as the remove command causes issues when citations are removed. Thank you for the suggestion.**

4. *just having the technical opportunity to setup $8x10^12$ model configuration doesn't actually seem to be a step forward as the vast majority of combinations are most likely not sensible ones. Thus, it seem to be very important to promote a tool like RavenR to guide users through the model setup.*

   **We certainly agree that tools to help guide this selection and provide starting points for new users in particular are important, and are happy to have some of this functionality in the package and paper.**

**3.2 Technical Comments**

1. *Fig 2 & 3: why are the referenced sections in bold font?*

   **We have removed the bold format for section references.**

2. *why is example code included and labeled as a figure? Wouldn't it be more straight forward to implement it as code blocks?*

   **I was unaware of the option to include the sample code as a code block format, but am happy to address this in the formatting stage prior to publication. Thank you for the recommendation.**

3. *Fig 5B: which actual variables are sim and obs? I assume all three of the others are forcing variables? Just to know that both curve are (probably) the same quantity without information about what they are, doesn't help with model evaluation.*

   **Thank you for the question, we have added to the caption of Figure 5 to clarify the variables in plot B, and added to line 359 to further clarify sim and obs.**